# Clinical Features and Novel Genetic Variants Associated with Hermansky-Pudlak Syndrome

**DOI:** 10.3390/genes13071283

**Published:** 2022-07-20

**Authors:** Chonglin Chen, Ruixin Wang, Yongguang Yuan, Jun Li, Xinping Yu

**Affiliations:** State Key Laboratory of Ophthalmology, Zhongshan Ophthalmic Center, Sun Yat-sen University, Guangdong Provincial Key Laboratory of Ophthalmology and Visual Science, Guangzhou 510060, China; chenchonglin@gzzoc.com (C.C.); wangruixin@gzzoc.com (R.W.); yuanyongguang@gzzoc.com (Y.Y.); lijun@gzzoc.com (J.L.)

**Keywords:** Hermansky-Pudlak syndrome, clinical characteristics, genetic characteristics

## Abstract

Hermansky-Pudlak syndrome (HPS) is a rare autosomal recessive syndromic form of albinism, characterized by oculocutaneous albinism (OCA) and other systemic complications. The purpose of this study was to investigate patients with HPS-associated gene mutations and describe associated ocular and extraocular phenotypes. Fifty-four probands clinically diagnosed as albinism were enrolled. Ophthalmic examinations and genetic testing were performed in all subjects. The phenotypic and genetic features were evaluated. HPS-associated gene mutation was identified in four of the patients with albinism phenotype. Clinically, photophobia, and nystagmus was detected in all (4/4) patients, and strabismus was found in one (1/4) patient. Fundus examination revealed fundus hypopigmentation and foveal hypoplasia in all (8/8) eyes. Eight novel causative mutations were detected in these four HPS probands. Five (62.5%, 5/8) of the mutations were nonsense, two of the mutations were missense (25%, 2/8), and one of the mutations was frameshift (12.5%, 1/8). All patients in our study carried compound heterozygous variants, and all these pathogenic variants were identified to be novel, with most (62.5%, 5/8) of the mutations being nonsense. Our results improved the understanding of clinical ocular features, and expanded the spectrum of known variants and the genetic background of HPS.

## 1. Introduction

Hermansky-Pudlak syndrome (HPS), first identified by Frantisek Hermansky and Paulus Pudlak in 1959, is a rare inherited disorder characterized by oculocutaneous albinism (OCA), bleeding diathesis, and/or comorbidities such as granulomatous colitis, neutropenia, and fatal pulmonary fibrosis [1,2,3]. The ophthalmic manifestation of HPS patients includes low visual ability, photophobia, nystagmus, strabismus, iris translucency, fundus hypopigmentation, and foveal hypoplasia [4,5,6,7,8].

HPS (MIM# 203300) is a genetically heterogeneous autosomal recessive multisystem disorder [3]. Up to now, eleven causative genes, including *HPS1*, *AP3B1*, *AP3D1*, *HPS3*, *HPS4*, *HPS5*, *HPS6*, *DTNBP1*, *BLOC1S3*, *BLOC1S6*, and *HPS11*, have been identified for HPS subtypes (HPS-1~HPS-11) [3,9,10,11]. These genes encode proteins that assemble a set of multi-protein complexes, each involved in steps of lysosome-related organelles biogenesis [10,12], and the aforementioned clinical manifestations are associated with defects in lysosome-related organelles, including melanosomes in melanocytes and delta granules in platelets [13]. Because of the low morbidity, genetic heterogeneity and phenotypic heterogeneity, no accurate correlations between HPS gene variants and clinical phenotype have been established.

Due to the low number of articles describing ophthalmic features of HPS, herein, our study mainly exhibited ophthalmic features in HPS and report eight novel variants in the *HPS3*, *HPS5* and *HPS6*. Our study expands the variant spectrum of HPS gene and highlights the importance of genetic testing for establishing an accurate diagnosis of HPS.

## 2. Patients and Methods

The study was approved by the Institutional Review Board of Zhongshan Ophthalmic Center, Sun Yat-sen University, adhered to the tenets of the Declaration of Helsinki, and all works were performed according to the approved study protocol. Informed written consent was obtained from the parents or guardians of each included proband, because all the participants were children under 18 years old.

Clinical data were collected, including age at presentation, gender, family history, and ocular findings in family members. Complete ophthalmic examination was performed in participants, including the best-corrected visual acuity, intraocular pressure, refractive errors, slit-lamp biomicroscopy, binocular indirect ophthalmoscopy, and front segment photography. To evaluate the fundus, digital fundus photography and optical coherence tomography (OCT) were performed.

Blood samples from the proband, the parents, as well as from their available family members were collected. Whole-exome sequencing was then performed. Genomic DNA was extracted using the TIANamp Blood DNA Kit (Tiangen Biotech, Beijing, China) according to the manufacturer’s instructions. The quantity and quality of DNA were verified with NanoDrop (2000c Model, Thermo Fisher, Waltham, MA, USA). Illumina paired-end libraries were prepared using the Kapa LTP library prep kit (Roche, Basel, Switzerland) according to the manufacturer’s protocol. To evaluate identified causative variants, the Human Gene Mutation Database and the gnomeAD, dbSNP and ExAC databases were consulted. To estimate the potential deleteriousness of the missense variants, the Mutation Taster, SIFT and PolyPhen2 tools were used. Identified mutations were then validated by Sanger sequencing of DNA from the patients and family members. All coding sequences and intron-exon junctions were amplified and sequenced comprehensively, and this was followed by co-segregation testing to verify suspected variants in the patients’ family members.

## 3. Results

### 3.1. Demographic and Clinical Features in Patients with HPS

Fifty-four probands clinical diagnosed as OCA according to the typical features were enrolled to perform genetic testing. Among these, HPS was identified in four probands (4/54, 7.4%). Two patients had a history of bleeding diathesis, but none was with pulmonary fibrosis or colitis. The demographic details and clinical features are summarized in Table 1.

### 3.2. Ocular Phenotypic Characteristics in Patients with HPS

A total of 8 eyes in 4 HPS participants were analyzed. The best-corrected visual acuity of the H1 proband was 0.5 (LogMAR) in each eye, 1.0 in both eyes of the proband H4, and not available in H2 and H3. Photophobia was observed in all (8/8) eyes, with the iris color being hazel, or brownish-black. Nystagmus was detected in all (4/4) patients and strabismus was found in one (1/4) patient. Fundus examination revealed fundus hypopigmentation and foveal hypoplasia in all (8/8) eyes. Ocular phenotypic characteristic of the patients with HPS were summarized in Table 2 and representative images of ocular manifestations in patients with HPS are shown in Figure 1.

### 3.3. Mutations Detected in Patients with HPS

Eight causative mutations were detected in the four HPS probands. Results revealed that H1 patient carried compound heterozygous variants in *HPS6* NM_024747, c.1021C>T (p.Gln341*) and c.1146_1147delTC (p.Gly382Glyfs*13). The H2 patient harbored a homozygous variant in *HPS3* NM_032383.5, c.15C>G (p.Tyr5*) and c.1838C>G (p.Ser613*). H3 and H4 patients carried compound heterozygous variants in *HPS5* NM_181507.2, c.1900G>T (p.Glu634*) and c.737G>A (p.Trp246*), and *HPS5* NM_181507.2, c.149G>A (p.Gly50Asp) and c.1078T>C (p.Cys360Arg), respectively. Pedigree analysis of the four families is shown in Figure 2.

Among the pathogenic variants, the coding impact of five (62.5%, 5/8) of the mutations was nonsense, two of the mutations were missense (25%, 2/8), and one of the mutations was frameshift (12.5%, 1/8). Eight (100%, 8/8) of the identified variants were confirmed to be novel and have not been reported previously. The details for causative mutations identified in patients with HPS are summarized in Table 3.

## 4. Discussion

Hermansky-Pudlak syndrome (HPS), a rare autosomal recessive syndromic form of albinism, is characterized by oculocutaneous albinism (OCA), bleeding diathesis, and serious complications including colitis, pulmonary fibrosis and immunodeficiency. The global prevalence of albinism is approximately 1:20,000, whereas that of HPS is only 1–9:1,000,000 individuals [14,15,16,17]. Previously, the diagnosis of HPS is primarily limited in clinical features and a platelet test shows an absence or severe reduction in platelet dense granules [12,18]. Thanks to recent advancements in genetic techniques, the diagnosis of different subtypes of HPS has become more precise and effective [19].

One interesting finding of this study is that only two probands had a history of bleeding diathesis, but none was with pulmonary fibrosis or colitis. Patients with HPS frequently suffer bleeding diathesis, granulomatous colitis, neutropenia, and fatal pulmonary fibrosis [2,3]. Patients in our study displayed relatively mild symptoms; on the one hand, these patients were all younger than 7 years old, and on the other hand, some symptoms of HPS do not manifest until the middle age and become fatal within a decade [20]. Due to the young age of the patients in our study, genotype–phenotype correlation is limited regarding the late-onset clinical comorbidities. Our results suggest timely examinations and lifelong monitoring should be conducted for these patients.

Another noteworthy finding of this study is that all the patients in our study carried compound heterozygous variants, and most (62.5%, 5/8) of the mutations were nonsense. Presently, the treatment for HPS is limited to supportive care, as there is no definitive treatment, and no drugs have been approved for HPS. To establish a reliable HPS diagnosis is crucial to identify HPS from OCA and formulate a long-term clinical plan. Our study emphasizes the importance of preemptive treatment, family planning, and personal plans for the future.

The limitations of our study are as follows. First, albinism is a rare disease, so there are fewer HPS patients, and the total number of HPS patients is limited. Second, as our center is a tertiary referral hospital for ocular diseases, referral bias could be present.

## 5. Conclusions

In conclusion, nystagmus, fundus hypopigmentation, and foveal hypoplasia were detected in all patients in our study, while pulmonary fibrosis or colitis was not observed because of the young age. Pathogenic mutations in HPS-associated genes were detected in all these four patients. All patients carried compound heterozygous variants, and all these pathogenic variants were identified to be novel, with most (62.5%, 5/8) of the mutations being nonsense. Our results improved understanding of clinical ocular features and expanded the spectrum of known variants and the genetic background of HPS. We suggest patients with albinism should receive timely examinations and genetic testing, and may also require pre-emptive treatment and family planning.

## Figures and Tables

**Figure 1 genes-13-01283-f001:**
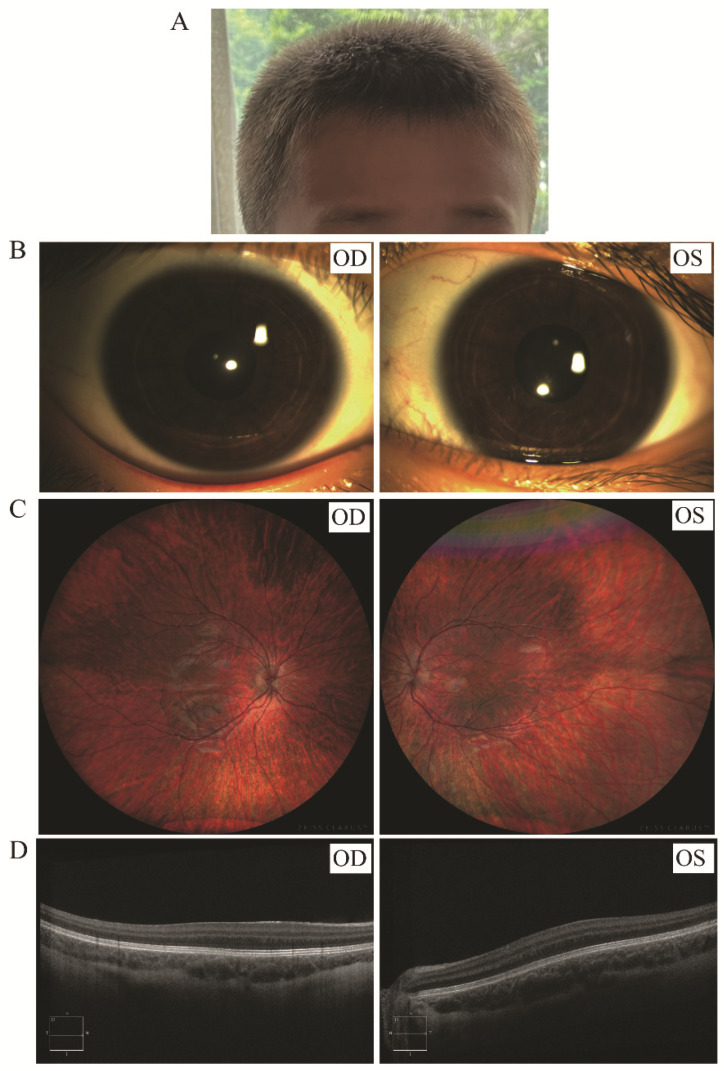
Representative clinical findings in patients with causative *HPS6* mutation. Figure shows a 5-year-old patient who carried compound heterozygous variants in *HPS6* NM_024747, c.1021C>T (p.Gln341*) and c.1146_1147delTC (p.Gly382Glyfs*13). His hair color is brown (**A**) and iris color is brownish-black (**B**). Hypopigmentation was detected in the fundus (**C**) and foveal hypoplasia was shown in the OCT (**D**).

**Figure 2 genes-13-01283-f002:**
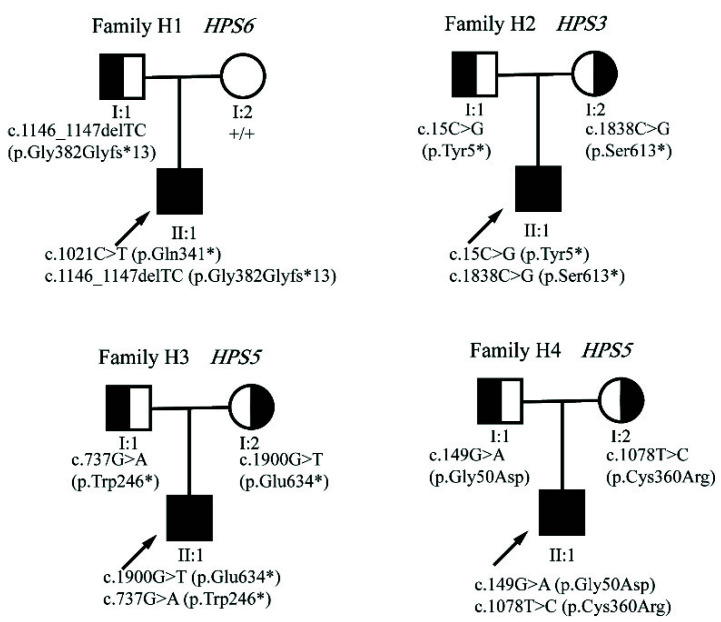
Schematic pedigrees of the families with causative HPS-associated gene mutations. Arrows indicate proband; filled symbols indicate compound heterozygous; Half-filled areas indicate heterozygous and unfilled indicate unaffected individuals.

**Table 1 genes-13-01283-t001:** Demographics and clinical findings of the HPS patients.

Patient ID	Gender	Age at Diagnosis	Skin Color	Hair Color	Bleeding Diathesis	Pulmonary Fibrosis	Colitis
H1	M	5 y	White	Brown	+	−	−
H2	F	7 month	White	Brownish-blonde	−	−	−
H3	F	6 month	White	Brownish-yellow	−	−	−
H4	F	6 y	White	Brownish-black	+	−	−

F, Female; M, Male; +, Positive; −, Negative.

**Table 2 genes-13-01283-t002:** Ocular features of the HPS patients.

Patient ID	BCVA (LogMAR) OD; OS	Photophobia	Iris Color	Nystagmus	Strabismus	Fundus Hypopigmentation	Foveal Hypoplasia
H1	0.5; 0.5	+	Brownish-black	+	−	+	+
H2	NA; NA	+	Hazel	+	−	+	+
H3	NA; NA	+	Hazel	+	−	+	+
H4	1.0; 1.0	+	Brownish-black	+	Esotropia	+	+

BCVA, Best corrected visual acuity; NA, Not available; +, Positive; −, Negative, OD, Right eye; OS, Left eye.

**Table 3 genes-13-01283-t003:** Causative mutations identified in the HPS patients.

Patient ID	Gene	cDNA Change	Protein Change	Coding Impact	Allele Frequency	Mutation Taster	SIFT	Polyphen2	ACMG	Source
H1	*HPS6*	c.1021C>T	p.Gln341*	Nonsense	NA	-	-	-	Likely Pathogenic	Novel
	*HPS6*	c.1146_1147delTC	p.Gly382Glyfs*13	Frameshift	NA	-	-	-	Likely Pathogenic	Novel
H2	*HPS3*	c.15C>G	p.Tyr5*	Nonsense	NA	-	-	-	Pathogenic	Novel
	*HPS3*	c.1838C>G	p.Ser613*	Nonsense	0.0000131	-	-	-	Pathogenic	Novel
H3	*HPS5*	c.1900G>T	p.Glu634*	Nonsense	NA	-	-	-	Likely Pathogenic	Novel
	*HPS5*	c.737G>A	p.Trp246*	Nonsense	NA	-	-	-	Pathogenic	Novel
H4	*HPS5*	c.149G>A	p.Gly50Asp	Missense	NA	Disease causing	Damaging	Damaging	Uncertain Significance	Novel
	*HPS5*	c.1078T>C	p.Cys360Arg	Missense	NA	Disease causing	Damaging	Damaging	Uncertain Significance	Novel

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
