# Peer review of "Clinical Features and Novel Genetic Variants Associated with Hermansky-Pudlak Syndrome"

_genes, 2022, doi:10.3390/genes13071283_

Round 1

Reviewer 1 Report

The manuscript by Chen et al. describes the analysis by whole-exome sequencing (WES) of a cohort of 54 children with a clinical diagnosis of albinism, in search of individuals affected with the rare recessive condition Hermansky-Pudlak syndrome (HPS), one of the classes of oculocutaneous albinism. The authors describe 8 novel causative mutations in 4 probands and describe the ophthalmologic features observed (foveal hypoplasia, nystagmus, and photophobia in all patient eyes, plus strabismus in one patient).

The manuscript is well organized and the research design is adequate. The authors must revise their manuscript to correct typos, missing ends of sentences and other minor spell problems. I have only the following concerns:

(1) The section describing the genetic analysis performed is unacceptably short. The authors must detail how WES was performed (kit, sequencer, depth, etc) and how data were analyzed (pipeline, kind of aligner, kind of variant caller, whether analysis was restricted just to the known HPS genes, etc). Did they find any individual with just one pathogenic variant in an HPS gene (suggesting a second pathogenic variant not seen in the WES analysis)?

(2) It would be helpful to categorize all the 8 novel variants according to the ACMG rules (see doi:10.1038/gim.2015.30) . Most likely they will be pathogenic or likely pathogenic, but the authors should report this.

(3) The segregation reported in family H1, with an apparent de novo mutation (p.Gln341*), must be confirmed. Did the authors verify that the mutation arose de novo by performing genotyping for genetic markers close to HPS6 and proved that indeed one of the alleles came from the mother? A false maternity is indeed a strange finding, but the de novo status of the variant must be verified formally.

Reviewer 2 Report

Chen et al. describe a case series of four patients with Hermansky-Pudlak Syndrome (HPS). This work improves on current knowledge of the prevalence and clinical features of HPS. 

The manuscript is very well-written, and the techniques and results are described very well and support the authors' conclusions.

I do have a few suggestions to improve readability. I have also taken liberty to suggest some English and grammar edits.

1. Minor English/grammar edit: Line 11-change "genes" to "gene"

2. Minor English/grammar edit: Line 12-change "clinical" to "clinically"

3. Minor English/grammar edit: Line 15-change "among" to "of"

4. Minor English/grammar edit: Line 15-add "and" before nystagmus

5. Minor English/grammar edit: Line 21-change "most of" to "most"

6. Minor English/grammar edit: end of Line 21 - change "was" to "being"

7. Minor English/grammar edit: Line 31-The references (i.e., "[1-3]") should NOT be superscript

8. Minor English/grammar edit: Line 33-Please change to "...and/or foveal hyperplasia."

9. Minor English/grammar edit: Line 44-Please change to "Due to the low number of..."

10. Minor English/grammar edit: Line 59 (Typo) - Please change to "...segment photography. To evaluate the ..."

11. Minor English/grammar edit: Line 72 - Remove indent before "3.1 Demographic..."

12. Minor English/grammar edit: Line 76 - I believe that "blonde" is more standard than "yellow"

13. Minor English/grammar edit: Lines 74-78 - This is a style issue, and I can go either way, but most journals do not like for authors to add a table that just repeats the text. I do not mind, but I would consider removing most of this text and just say something like "Demographics and clinical findings are summarized in Table 1." then keep table.

14. MAJOR: Lines 97-98 - Please change "color of hair..." to "iris color"

15. MAJOR: Line 110 - Please add "D" to end of sentence.

16. Minor English/grammar edit: Line 139 - Please remove extra space (change "1:20, 000" to "1:20,000").

17. Minor English/grammar edit: Line 142 - Please change "Thanks for..." to "Thanks to..."

18. Minor English/grammar edit: Line 144 - Please change "One of the interesting..." to "One interesting..."

19. Minor English/grammar edit: Line 154 - Please change "...mutations was non-..." to "...mutations were non-..."

20. Minor English/grammar edit: Line 164 - Please add a comma after "hyperpigmentation"

21. Minor English/grammar edit: Line 169 - Please change "...was nonsense..." to "...being nonsense..."

22. Minor English/grammar edit: Line 170 - Highly suggest changing "...We suggested..." to "...We suggest..."

23. References all need checked for proper style according to MDPI guidelines.
